

# Biological properties of mucus from land snails (*Lissachatina fulica*) and freshwater snails (*Pomacea canaliculata*) and histochemical study of mucous cells in their foot

Phornphan Phrompanya[1,2], Narinnida Suriyaruean[1], Nattawadee Nantarat[1], Supap Saenphet[1], Yingmanee Tragoolpua[1] and Kanokporn Saenphet[1]

[1] Department of Biology, Faculty of Science, Chiang Mai University, Chiang Mai, Thailand
[2] Ph.D.'s Degree Program in Biology (International Program), Faculty of Science, Chiang Mai University, Chiang Mai, Thailand

Corresponding author
Kanokporn Saenphet,
stit.lilo123@gmail.com

## ABSTRACT

**Background.** Mucus derived from many land snails has been extensively utilised in medicine and cosmetics, but some biological activities of the mucus need to be well documented. Nevertheless, most mucus is obtained from land snails, while mucus from freshwater snails has yet to be attended.

**Methods.** This study aims to determine and compare mucus's antioxidant and anti-inflammatory activities from the land snail *Lissachatina fulica* and the freshwater snail *Pomacea canaliculata*. ABTS, DPPH, reducing power and total antioxidant activity assays were used to evaluate the antioxidant capacity. Inhibition of nitric oxide production in lipopolysaccharide-activated RAW 264.7 cells was performed to determine the anti-inflammatory activity. Additionally, the histochemical analysis of mucous cells in each snail foot was conducted to compare the distribution of mucous cells and types of mucins using periodic acid-Schiff and Alcian blue staining.

**Results.** Mucus from *L. fulica* and *P. canaliculata* exhibited antioxidant and anti-inflammatory activities in different parameters. *L. fulica* mucus has higher total antioxidant ($44.71 \pm 2.11$ mg AAE/g) and nitric oxide inhibitory activities ($IC_{50} = 9.67 \pm 0.31\,\mu g/ml$), whereas *P. canaliculata* mucus has better-reducing power activity ($43.63 \pm 2.47$ mg AAE/g) and protein denaturation inhibition ($IC_{50} = 0.60 \pm 0.03$ mg/ml). Histochemically, both species' dorsal and ventral foot regions contained neutral and acid mucins in different quantities. In the dorsal region, the neutral mucins level in *L. fulica* ($16.64 \pm 3.46\%$) was significantly higher than that in *P. canaliculata* ($11.19 \pm 1.50\%$), while the acid mucins level showed no significant difference between species. Levels of both mucins in the ventral foot region of *L. fulica* ($15.08 \pm 3.97\%$ and $10.76 \pm 3.00\%$, respectively) were significantly higher than those of *P. canaliculata* ($2.25 \pm 0.48\%$ and $2.71 \pm 0.56\%$, respectively). This study revealed scientific evidence of the biological capacity of mucus from *L. fulica* and *P. canaliculata* as well as provided helpful information on the region of the foot which produces effective mucus.

## INTRODUCTION

Skin acts as a protective barrier against external threats such as UV radiation, pollutants, and pathogens. However, the skin is constantly exposed to these harmful factors, which can induce oxidative stress. An excess of reactive oxygen species (ROS), including free radicals, in the body can lead to cellular damage and dysfunction. Oxidative stress has been implicated in various health issues and skin problems such as wrinkling, fine lines, allergies and cancer (*Sander et al., 2003*; *Tsuchida & Kobayashi, 2020*). Moreover, oxidative stress can trigger an inflammatory response in the skin. Although inflammation is a natural defence mechanism, chronic or excessive inflammation can contribute to skin disorders such as acne, psoriasis, eczema, and rosacea (*Yang et al., 2022*). The search for effective antioxidants has gained considerable interest. Natural substances are the main source of antioxidants, including vitamins, carotenoids, and phenolic compounds from plants, as well as animal proteins and peptides. These antioxidants are considered safe products. Consequently, plant- and animal-based antioxidants are widely applied in the pharmaceutical and cosmetic industries (*Abeyrathne et al., 2022*).

Snail secretion, or snail mucus, is one of the most popular ingredients in cosmeceutical products. The mucus is secreted by mucous glands located in the footplate and covers the whole external surface of the animal. It is mainly used to reduce friction, protect the snail foot during locomotion, maintain moisture, and help their mating and hunting activities (*Richter, 1980*). Snail mucus is a natural substance containing different biological properties, such as antimicrobial, anticancer, anti-inflammation, and wound healing (*Mane et al., 2021*; *McDermott et al., 2021*). Nowadays, various companies extensively use snail mucus in skin care products, and it is a growing market (*McDermott et al., 2021*). *Lissachatina fulica*, the giant African snail, is a terrestrial snail whose mucus has been used since ancient times. Aqueous extracts of mucus from *L. fulica* showed *in vitro* anti-inflammatory activities, including anti-proteinase, anti-lipoxygenase and anti-protein denaturation (*Wiya, Nantarat & Saenphet, 2020*), antibacterial, and wound healing activities (*Santana et al., 2012*). Mucus from *L. fulica* is applied as a cosmetic ingredient (*Nguyen, Masub & Jagdeo, 2020*), but there still needs to be more reporting on some biological properties of this mucus.

Despite the high species diversity of snails, the mucus from only a few species has been studied and applied in commercial products. *Pomacea canaliculata*, the golden apple snail, is a compelling freshwater snail that has rich mucus covering the body surface. The antibacterial activity of the mucus from this species has been reported (*Nantarat, Tragoolpua & Gunama, 2019*), but its antioxidant and anti-inflammatory activities have yet to be evaluated.

Mucins, the bioactive substances in mucus, are classified into neutral or acid mucins, distinguished based on their histochemical characteristics (*Filipe, 1979*). Acid mucins are

quite important as a protective function due to their moisturising and antibacterial abilities (*Faillard & Schauer, 1972*; *Cao & Wang, 2009*). Therefore, mucus that contains a high acid mucin proportion should provide better biological activity and cosmeceutical properties. Several studies used histochemical staining techniques to classify mucin-type, associated with function in each organ (*Grau et al., 1992*; *Sarasquete et al., 2001*; *Greistorfer et al., 2017*). The foot histology of *L. fulica* and *P. canaliculata* was demonstrated in our previous study, which found differences of mucous cell distribution between the dorsal-ventral foot area and between species (*Phrompanya et al., 2022*). The mucin types and mucous cell distribution of *L. fulica* foot tissue were investigated only at the juvenile stage (1–3 months) using histological and histochemical techniques (*Suwannapan et al., 2019*). Nevertheless, a histochemical study of the *P. canaliculata* foot has not been presented, only a basic histological study was revealed (*Peña & Pocsidio, 2017*).

Wide commercial products incorporate mucus derived from *L. fulica*, but some biological effects and the foot histochemistry of this species have yet to be well documented. Moreover, a high mucous-secreting snail, like *P. canaliculata*, does not get as much attention. Therefore, the present study proposes to evaluate and compare the biological activities, such as the terrestrial and freshwater snail's antioxidant and anti-inflammatory effects. In addition, this study investigated their foot histochemical features, focusing on the mucous cells to classify types of mucus in different regions of the foot and compare these two species. The results provided more scientific evidence of the biological properties of the snail mucus and the histochemical pattern of the mucus-secreting area in the snail foot, which helps obtain an effective mucus during collection.

## MATERIALS & METHODS

### Animals and sample collection

Thirty adult giant African snails (*Lissachatina fulica*) and golden apple snails (*Pomacea canaliculata*) were collected from Chiang Mai province and identified based on shell morphology and compared with original descriptions (*Brandt, 1974*). The collected snails were grown in the laboratory. All animal procedures were approved by the Ethics and Animal Care Committee of Chiang Mai University, following the guidelines given by the National Institute of Health Guide for the Care and Use of Laboratory Animals.

### Mucus collection

To avoid contamination, the snails were kept without feeding for three days before mucus collection. Subsequently, the snails were manually stimulated at their foot regions and approximately 2 ml of mucus secretion per individual were collected, filtered through Whatman No. 1 filter paper using vacuum filtration, and stored at 4 °C. The filtered mucus was analyzed for total protein content using the method from *Bradford (1976)* before *in vitro* analysis.

### Chemicals and reagents

All chemicals were analytical reagent grade. Gallic acid, 1,1-diphenyl-2-picrylhydrazyl (DPPH), 2,2-azino-bis-(3-ethylbenzothiozoline-6-sulphonic acid) (ABTS), hematoxylin,

eosin, alcian blue 8GX (AB), sulfanilamide, N-naphthylethylenediamine dihydrochloride, Dulbecco's Modified Eagle Medium (DMEM) and 3-(4,5-Dimethylthiazol-2-yl)-2,5-diphenyltetrazolium bromide (MTT) were purchased from Sigma-Aldrich (St. Louis, MO, USA). Methanol, trichloroacetic acid, sulfuric acid, deionised water, dimethyl sulfoxide (DMSO), formaldehyde, hydrochloric acid (HCl), sodium sulphate, potassium persulfate and potassium ferricyanide were purchased from RCL Labscan (Bangkok, Thailand). Ascorbic acid, ferric chloride and ammonium molybdate were purchased from Ajax Finechem Pty. Ltd. (Auckland, New Zealand).

## Determination of *in vitro* antioxidant activity

The antioxidant activity assays were performed to evaluate the capacity of snail mucus to inhibit different free radicals DPPH, and ABTS radical scavenging assays were conducted based on hydrogen atom transfer and electron transfer reactions, respectively. Reducing power ability was evaluated by measuring the transformation of Fe (III) to Fe (II), and total antioxidant activity was evaluated by measuring the reduction of Mo (VI) to Mo (V).

### DPPH radical scavenging activity

The DPPH radical scavenging activity of mucus and gallic acid (used as a standard) was determined in terms of hydrogen-donating ability following the method of *Tagashira & Ohtake (1998)*. The stable DPPH radical solution was freshly prepared at 0.004% in methanol. A total of 100 µl of the sample were mixed with 3.0 ml of DPPH solution and incubated for 30 min in the dark. The absorbance was measured at 517 nm. The scavenging activity was calculated based on the percentage of DPPH scavenged using the following formula:

$$\% \text{ scavenging} = [(A_0 - A_s)/A_0] \times 100$$

Where $A_0$ is the absorbance of the DPPH solution and $A_s$ is the absorbance of the sample at 30 min (mucus or gallic acid).

### ABTS radical scavenging activity

The ABTS assay was conducted according to the method of *Re et al. (1999)*. The ABTS radical stock solution was prepared by mixing equal volumes of 7 mM ABTS solution and 2.45 mM potassium persulfate solution. The solution was incubated for 12 h in the dark to create ABTS radicals observed by the solution, which was turned to dark. This solution was freshly diluted using deionised water to create a working solution before each assay for an initial absorbance of about $0.7 \pm 0.02$ at 734 nm. The radical scavenging assay was performed by adding 20 µl of different concentrations of mucus to 2.0 ml of ABTS working solution. After mixing the solution, the decrease in absorbance was read after 1 min. The absorbance was recorded at 734 nm against distilled water and compared with the ABTS solution. Gallic acid was used as the positive control. The activity was evaluated as the percentage of ABTS radicals scavenged using the following formula:

$$\% \text{ scavenging} = [(A_0 - A_s)/A_0] \times 100$$

Where $A_0$ is the absorbance of the ABTS solution and $A_s$ is the absorbance of the sample (mucus or gallic acid).

### Reducing power ability

The reducing ability of samples was estimated using a Fe (III) to Fe (II) reduction assay (*Oyaizu, 1986*). 100 µl of different samples and ascorbic acid (used as a standard) were added to 1.0 ml of phosphate buffer (0.2 M, pH 6.6) and 1.0 ml of potassium ferricyanide (1% w/v). The mixtures were incubated at 50 °C for 20 min, then 1.0 ml of 10% trichloroacetic acid was added to each tube. 2 ml of the mixture was mixed with an equal volume of distilled water and 0.2 ml of ferric chloride solution (0.1% w/v). The absorbance was measured at 700 nm, and phosphate buffer was used as the blank solution.

### Total antioxidant activity

The total antioxidant activity of samples was determined using the phosphomolybdate assay (*Umamaheswari & Chatterjee, 2008*) based on the Mo (VI)-Mo (V) reduction of the sample. A total of 3.0 ml of the reaction mixture, containing 0.6 M sulfuric acid, 28 mM sodium phosphate, and 4 mM ammonium molybdate, was mixed with 300 µl of sample and incubated at 95 °C for 90 min. After cooling, the absorbance of the solution was read at 695 nm against distilled water as the blank. The total antioxidant capacity was presented as mg ascorbic acid equivalent per g protein (mg AAE/g protein).

## Determination of anti-inflammatory activity

The anti-inflammatory activity was evaluated by measuring the inhibition of nitric oxide (NO) production using a macrophage inflammatory assay and the inhibition of protein denaturation.

### Cell culture and cell viability assay

RAW 264.7, a mouse macrophage cell line, was obtained from the Division of Microbiology, Department of Biology, Faculty of Science, Chiang Mai University, Chiang Mai, Thailand. The cells were cultured in DMEM containing 10% fetal bovine serum (FBS), 100 U/ml penicillin, and 100 µg/ml streptomycin at 37 °C in a 5% $CO_2$ humidified incubator. The cell viability was performed using a modified MTT assay (*Mosmann, 1983*). After overnight culture in a 96-well plate ($1 \times 10^6$ cells/well), the medium was discarded and replaced with 100 µl of the sample (0.4 to 200 µg protein/ml) then the cells were re-incubated for 24 h at 37 °C in 5% $CO_2$. After incubating, the sample in each well was removed, and 30 µl of MTT solution (2 mg/ml in phosphate-buffered saline) was added to each well and incubated at 37 °C for 4 h. Subsequently, 200 µl of DMSO was added to each well to dissolve the formazan crystals, and the absorbance was measured at 540 nm. The concentration of mucus that could reduce the absorbance to 50% compared to untreated cells was used as the median lethal concentration ($LC_{50}$) value.

### Inhibition of NO production

The level of NO, an acute inflammatory mediator released from macrophage cells, was indirectly determined based on the Griess reaction. RAW 264.7 cells ($1 \times 10^6$ cells/ml) were incubated with lipopolysaccharide (LPS) (1 µg/ml) and treated with 100 µl of mucus for 24 h. The amount of NO was indirectly measured in the form of nitrite in the culture medium. Briefly, the culture medium was incubated with an equal volume of Griess reagent for 15

min in the dark at room temperature, and the absorbance was measured at 550 nm. The inhibition of NO production was calculated as a percentage using the following formula, and each sample's half-maximal inhibitory concentration ($IC_{50}$) was determined.

$$\% \text{ inhibition} = [(A_c - A_s)/A_c] \times 100$$

Where $A_c$ is the absorbance of the control LPS and $A_s$ is the absorbance of the sample.

### Inhibition of protein denaturation assay

The method was conducted according to *Kumari et al. (2015)*. The reaction mixture consisted of 0.1 ml of egg albumin, 1.9 ml of phosphate-buffered saline (PBS, pH 6.4), and 1 ml of the mucus at different concentrations. Distilled water (1 ml) was used instead of the mucus as a negative control. The reaction mixture was incubated at 37 °C for 15 min and then heated at 70 °C for 5 min. After cooling at room temperature, the absorbance was measured at 660 nm using a spectrophotometer. Sodium diclofenac was used as a standard drug in concentrations of 0.01, 0.1, 0.25, 0.5, and 1.0 mg/ml. Inhibition of albumin denaturation was calculated as a percentage using the following equation, and the half-maximal inhibitory concentration ($IC_{50}$) of each sample was determined.

$$\% \text{ inhibition} = [(A_s - A_c)/A_c] \times 100$$

Where $A_c$ is the absorbance of control and $A_s$ is the absorbance of the sample.

## Histochemical analysis

### Tissue sample preparation

Ten adults of each snail species were used to prepare tissue sections. Small pieces of the foot were dissected and fixed in 10% neutral buffered formalin for 24 h. The fixed tissues were decalcified by soaking in 10% HCl for 2 h and neutralised by sodium sulphate for 12 h. The tissues were washed in running tap water before processing using a standard paraffin technique and cut into 6 μm-thick sections for histological and histochemical study.

### Tissue staining

The sections from every tissue sample were stained with Harris's haematoxylin and eosin (H&E) for standard histological study (*Harris, 1900*). For histochemical characterisation, sections were stained with periodic acid-Schiff (PAS) for the detection of neutral mucins (*McManus, 1948*) and Alcian blue (AB) at pH 2.5 for acid mucins (*Mowry, 1956*). All sections were observed under a light microscope and photographed using an Olympus DP12 camera.

### Quantification of mucins

The PAS- and AB-stained sections were used to identify the different types of mucin-secreting mucous cells in the dorsal and ventral foot surfaces. Images of each surface were taken (three fields per section and three sections per individual). Image analysis software, ImagesJ (National Institutes of Health, version 1.50i), was used to measure the area of stained mucin following the colour deconvolution and quantification method (*Ruifrok & Johnston, 2001*). The results were calculated as a percentage area of acid or neutral mucins and presented as mean ± SD values.

**Table 1** Antioxidant activity of mucus from *L. fulica* and *P. canaliculata*.

| Mucus (1 mg/ml) | DPPH (% inhibition) | ABTS (% inhibition) | Reducing power (mg AAE/g) | Total antioxidant (mg AAE/g) |
|---|---|---|---|---|
| *L. fulica* | NA | $2.80 \pm 1.69^b$ | $36.02 \pm 3.21^a$ | $44.71 \pm 2.11^a$ |
| *P. canaliculata* | NA | $1.63 \pm 0.90^b$ | $43.63 \pm 2.47^b$ | $33.27 \pm 2.59^b$ |
| Gallic acid (0.1 mg/ml) | $92.75 \pm 0.37$ | $79.80 \pm 1.34^a$ | – | – |

Notes.
Data are represented as the mean $\pm$ SD of three measurements. Different letters (a and b) for each column indicate significant differences ($p < 0.05$). NA, not available.

## Statistical analysis

The results were analysed using SPSS version 17 for Windows. For antioxidant and anti-inflammatory activities, independent sample t-tests and one-way ANOVA, followed by Tukey's test for multiple comparisons, were used to compare the ability between mucus of different species. For the study of mucin quantification, an independent sample $t$-test was used to compare the percentage area of mucin between dorsal-ventral areas and between different species at a 5% level of significance.

## RESULTS

Protein concentration in the filtered fresh mucus was estimated before use. The results showed that the protein contents of the *L. fulica* mucus and the *P. canaliculata* mucus were $1.49 \pm 0.37$ and $1.13 \pm 0.19$ mg/ml, respectively. Therefore, the mucus concentration was adjusted to 1 mg/ml, which was used as the maximum concentration for this study.

### Antioxidant activity

Antioxidant activities of the snail mucus are shown in Table 1. Mucus from *L. fulica* and *P. canaliculata* at 1 mg/ml showed no free radical scavenging activity of DPPH and weakly scavenged ABTS radical cation. The *L. fulica* mucus exhibited the reducing power of 36.02 mg AAE/g protein, while *P. canaliculata* mucus showed a higher reducing power of 43.63 mg AAE/g protein at the same concentration. Furthermore, the result of total antioxidant activity revealed that the *L. fulica* mucus showed a higher total antioxidant activity of 44.71 mg AAE/g.

### Anti-inflammatory activity

Cell viability and NO inhibitory activity of the mucus are shown in Table 2 as the $LC_{50}$ value for cytotoxicity and the $IC_{50}$ value for inhibition of NO production. The concentrations of the mucus that yielded cell viability higher than 90% were used for NO inhibitory analysis. It was observed that 90% of the RAW 264.7 macrophage cells were viable at the concentration of 0.39 to 12.5 µg/ml for *L. fulica* mucus, and the $LC_{50}$ value is 96.89 µg/ml. The percent viability of the RAW 264.7 cells treated with *P. canaliculata* mucus was lower than that treated with *L. fulica* mucus at the same concentrations with a non-significant difference (Fig. 1). The highest concentrations of the mucus from *L. fulica* and *P. canaliculata* used in the NO production assay were 12.50 and 0.78 µg/ml, respectively.

**Table 2** Cytotoxic activity and nitric oxide inhibitory activity of mucus from *L. fulica* and *P. canaliculata*.

| Mucus | *L. fulica* | *P. canaliculata* |
| --- | --- | --- |
| Cytotoxic activity ($LC_{50}$ [µg/ml]) | 96.89 ± 12.40 | 102.86 ± 10.70 |
| Nitric oxide inhibitory activity ($IC_{50}$ [µg/ml]) | 9.67 ± 0.31 | – |

**Notes.**
Data are represented as the mean ± SD of three measurements.

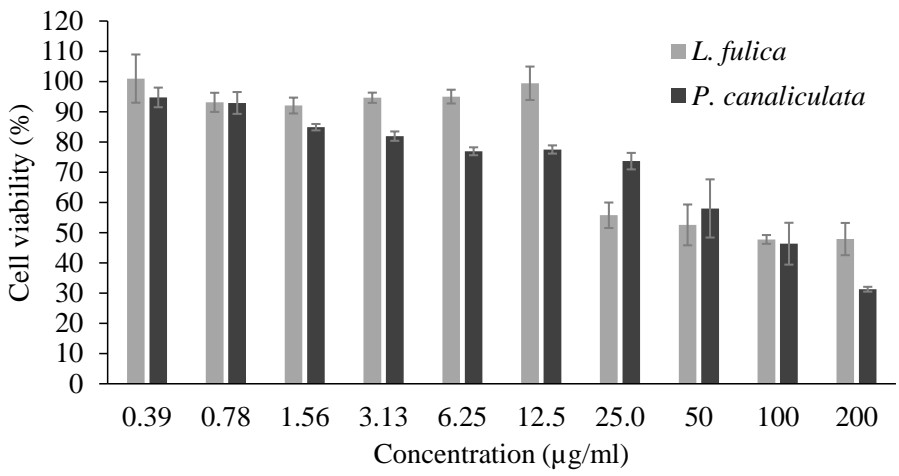

**Figure 1** The percentage viability of RAW 264.7 cells treated with various concentrations of mucus from *L. fulica* and *P. canaliculata* for 24 h. The viability of cells was determined by MTT assay. The data were presented as mean ± SD of three independent experiments.

The inhibitory potentials of the mucus on proinflammatory molecule NO are presented in Fig. 2. The highest concentration of *P. canaliculata* mucus (0.78 µg/ml) exhibited only 16.79% NO inhibition; therefore, the $IC_{50}$ of *P. canaliculata* mucus could not be determined. *L. fulica* mucus exhibited a significant dose-dependent decrease in NO production. The highest concentration of *L. fulica* mucus (12.50 µg/ml) showed 61.90% NO inhibition, and its $IC_{50}$ value was 9.67 ± 0.31 µg/ml. Strong inhibitory activity was found in *L. fulica* mucus at 3.13, 6.25, and 12.5 µg/ml concentrations, presenting a significant difference from the untreated group (control LPS) at $p < 0.01$.

For the evaluation of protein denaturation, *L. fulica* and *P. canaliculate* exhibited the inhibitory capacity of 66.28% and 71.56%, respectively (Fig. 3). *P. canaliculata* mucus demonstrated a significant ability to inhibit protein denaturation compared to that of *L. fulica*, which the $IC_{50}$ values of each sample could indicate in Table 3.

## Histochemical characteristics of mucous cells in foot tissues

The histological structures of *L. fulica* and *P. canaliculata* showed that their foot tissues consist of two main layers: the epithelium, which contributes simple columnar cells, and the sub-epithelium, the location of most mucous cells. The results showed that a large number of mucous cells was located in the sub-epithelial layer of the dorsal foot areas of both species. The ventral foot area of *L. fulica* contained small mucous cells distributed

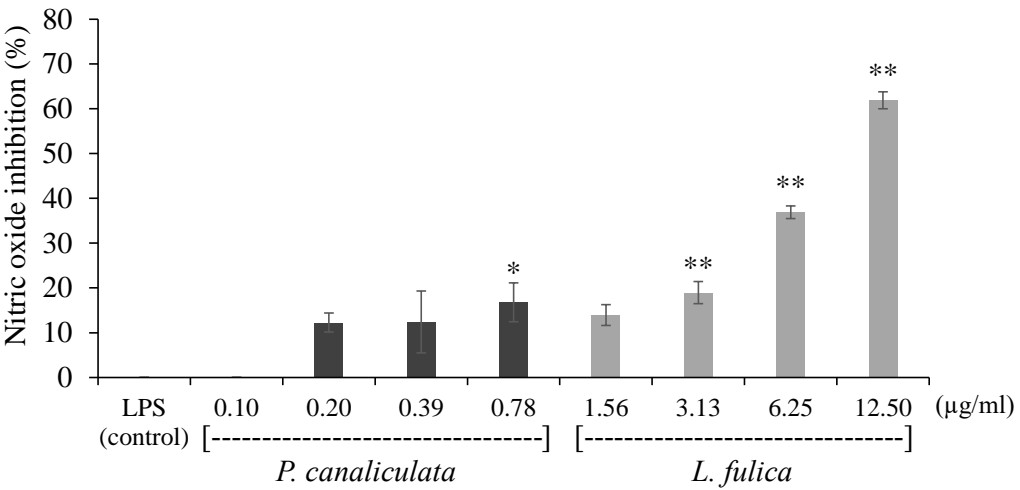

**Figure 2** **The inhibition of nitric oxide production in RAW 264.7 cells treated with non-toxic concentration of *P. canaliculata* mucus (0.10–0.78 μg/ml) and *L. fulica* mucus (1.56–12.50 μg/ml).** The data were represented as mean ± SD of three independent experiments. * and ** indicate significant statistical differences with the control at $p < 0.05$ and $p < 0.01$, respectively.

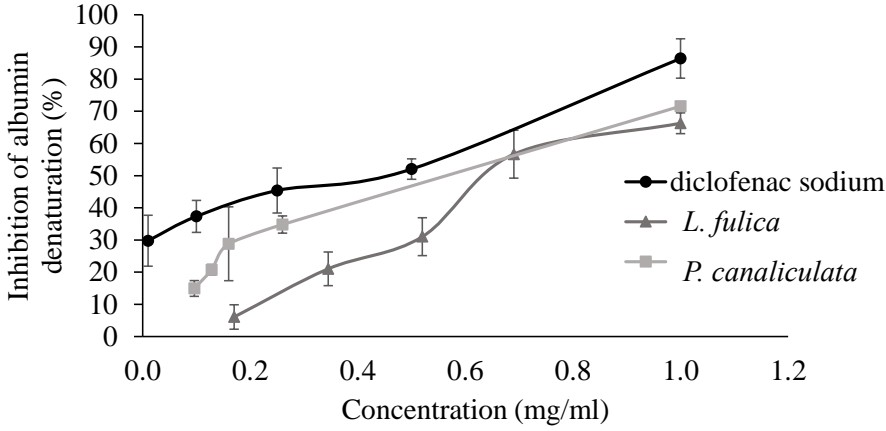

**Figure 3** **Effect of diclofenac sodium and mucus from *L. fulica* and *P. canaliculata* on albumin denaturation.** The data were presented as mean ± SD of percent inhibition of three independent experiments.

in the sub-epithelium. In contrast, mucous cells in the ventral foot area of *P. canaliculata* were found only in the epithelial layer.

The present study used special staining techniques to compare the histochemical characteristics of the foot tissues of two species of snails, focusing on mucous secretory cells in different regions of the foot. Two shapes of mucous cells, tubular and round, were found in the dorsal foot area of both species (Figs. 4A, 5A) in different sizes and levels of mucin histochemical reaction (Table 4). The mucous cells of both species were positively stained for PAS with a purple colour and AB (pH 2.5) with a blue colour, suggesting the

**Table 3** Inhibition of albumin denaturation of diclofenac standard drug and mucus from *L. fulica* and *P. canaliculata* expressed as $IC_{50}$ (mg/ml).

| Samples | Inhibition of albumin denaturation ($IC_{50}$ [mg/ml]) |
| --- | --- |
| *L. fulica* mucus | $0.74 \pm 0.05$[c] |
| *P. canaliculata* mucus | $0.60 \pm 0.03$[b] |
| Diclofenac sodium | $0.38 \pm 0.07$[a] |

**Notes.**
Data are represented as the mean $\pm$ SD of three measurements. Different letters (a–c) indicate significant differences ($p < 0.05$).

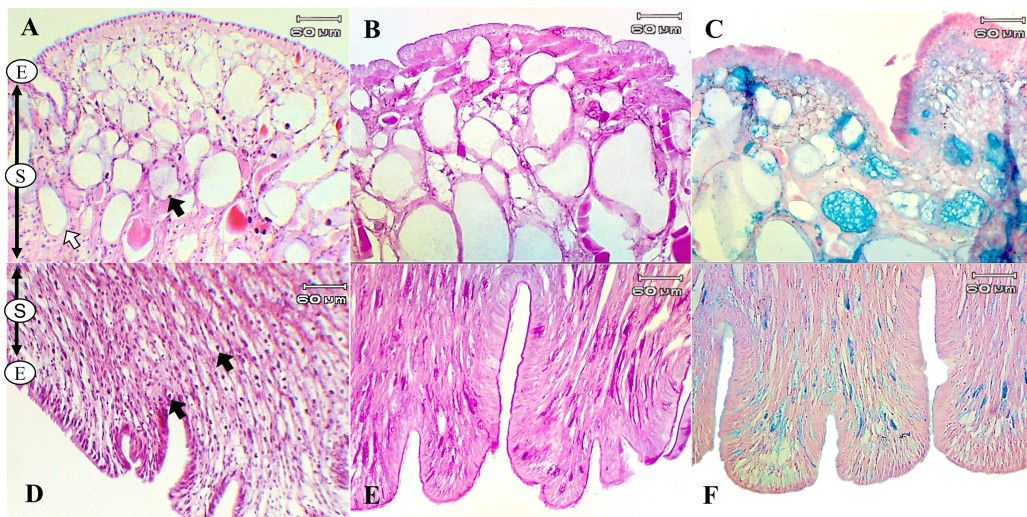

**Figure 4** **Histological and histochemical features of mucous cells in the *L. fulica* foot.** (A) The dorsal foot area with H&E stain showing epithelium (E) and tubular- (white arrow) and round-shaped (black arrow) mucous cells in the sub-epithelium (S). (B) The dorsal foot area with PAS stain. (C) The dorsal foot area with AB stain. (D) The ventral foot area with H&E stain containing packed mucous cells in the sub-epithelium (black arrow). (E) The ventral foot area with PAS stain. (F) The ventral foot area with AB stain.

presence of neutral and acid mucins, respectively (Figs. 4B–4C, 5B–5C). In *L. fulica*, large and medium mucous cells were densely located in the dorsal foot area with moderate PAS- and AB-positive reactions. Small mucous cells were scattered throughout the ventral area with the same reaction. In *P. canaliculata*, the dorsal foot area contained medium mucous cells with strong PAS- and AB-positive reactions, and the ventral area showed a moderate reaction with both stains in mucous cells in the epithelium.

This study also determined the quantification of the mucous layer by measuring the stained mucin area. The results determined that the dorsal region of the snail foot was the main area of mucus secretion related with a percentage of stained mucin areas. In particular, the neutral and acid mucins in the dorsal foot region of *P. canaliculata* had a significantly higher percentage of the stained area than that of the ventral region. Differences of the stained mucin areas were also found between species. The area of neutral mucins in the *L.*

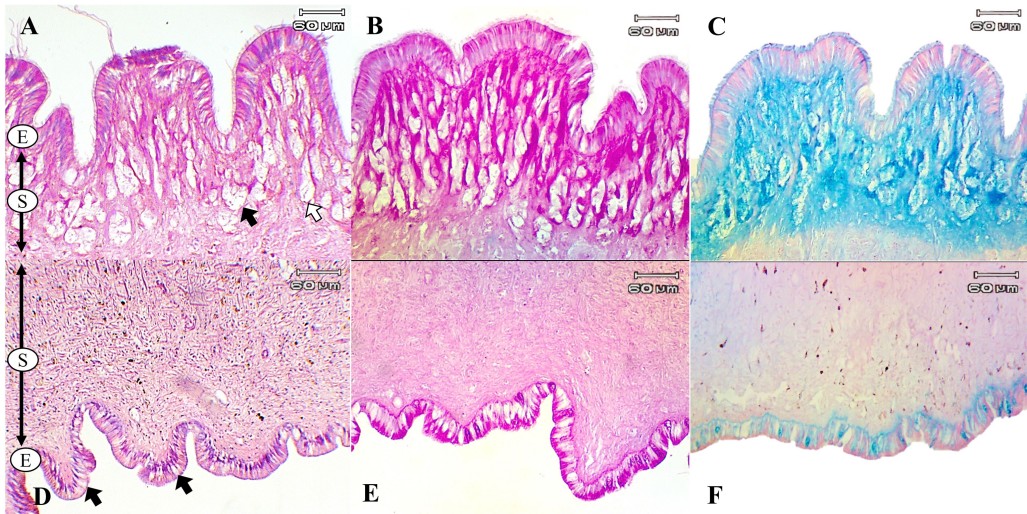

**Figure 5** **Histological and histochemical features of mucous cells in the *P. canaliculata* foot.** The dorsal foot area with H&E stain showing the epithelium (E) and tubular- (white arrow) and round-shaped (black arrow) mucous cells in the sub-epithelium (S). (B) The dorsal foot area with PAS stain. (C) The dorsal foot area with AB stain. (D) The ventral foot area with H&E stain showing small mucous cells in the epithelium (black arrow). (E) The ventral foot area with PAS stain. (F) The ventral foot area with AB stain.

**Table 4** **Mucous cells and mucin histochemistry of the secretions in the foot of *L. fulica* and *P. canaliculate*.**

| Foot region | Mucous cell shape | Cell size | PAS | AB (pH 2.5) | Types of mucins detected |
|---|---|---|---|---|---|
| ***L. fulica*** | | | | | |
| Dorsal area | Tubular cell | Large | ++ | + + | Neutral, acid |
| | Round cell | Large, medium | | | |
| Ventral area | Mucous cell | Small | + + | + + | Neutral, acid |
| ***P. canaliculata*** | | | | | |
| Dorsal area | Tubular cell | Medium | + + + | + + + | Neutral, acid |
| | Round cell | Medium | | | |
| Ventral area | Mucous cell | Small | + + | + + | Neutral, acid |

**Notes.**
+, indicates a weak positive reaction' + +, indicates a moderate positive reaction, and + + +, indicates a strong positive reaction.

*fulica* foot was significantly larger than that of *P. canaliculata* when compared to the same foot region. Acid mucins in the ventral foot region of *L. fulica* covered a significantly larger area than that in *P. canaliculata*. Still, there was no significant difference in the dorsal foot region between *P. canaliculata* and *L. fulica*. The area of stained mucin is shown in Fig. 6.

## DISCUSSION

Snail mucus is commonly used for treating injuries and as a key ingredient in cosmeceuticals. This is because several substances found in snail mucus have therapeutic properties that

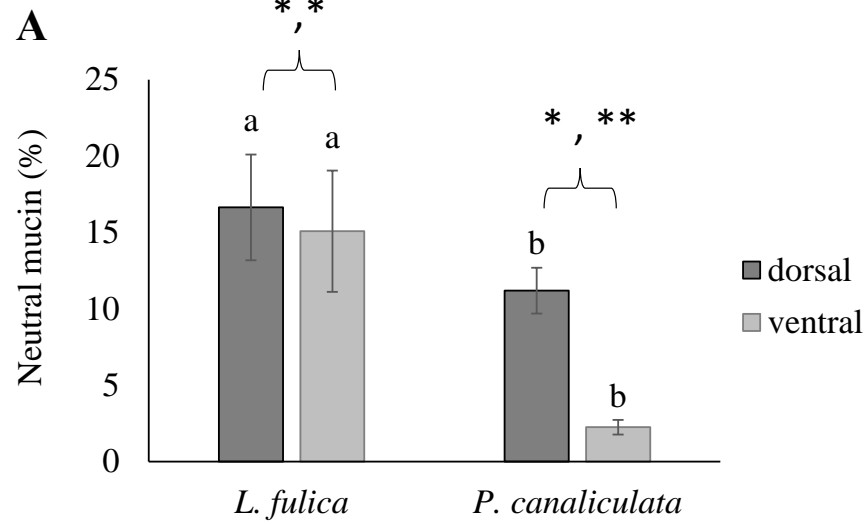

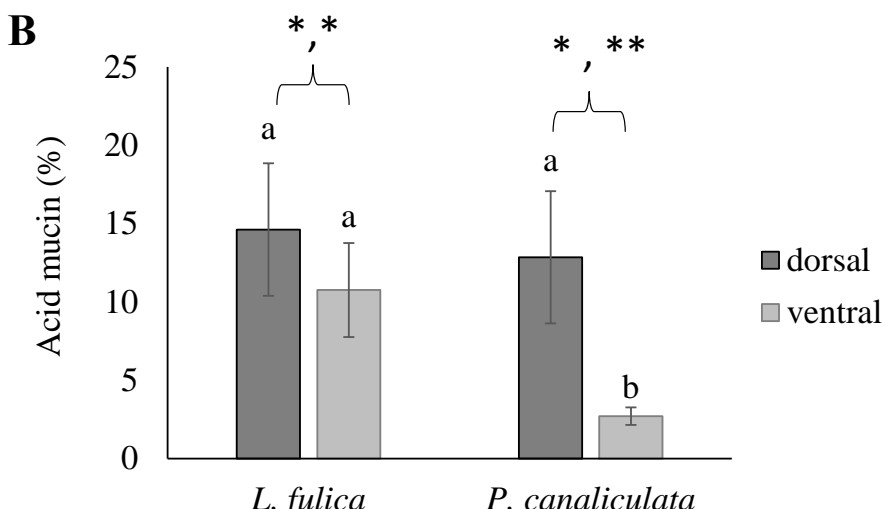

**Figure 6 Area of mucin secretion.** (A) Neutral mucin secretion. (B) Acid mucin secretion in the dorsal and ventral regions of the foot. Different letters above the bars indicate a significant difference between species ($p < 0.05$), and different numbers of asterisks above the bars indicate a significant difference between areas (*$p < 0.05$, ** $p < 0.01$).

provide antioxidant and anti-inflammatory benefits. This study determined the antioxidant ability of mucus from *L. fulica* and *P. canaliculata* using different free radical-generating systems.

The radical scavenging activity of mucus was evaluated by DPPH and ABTS assays. In the DPPH assay, the antioxidant activity of mucus from both species could not be exhibited. The mucus could scavenge the ABTS radical, indicating the ability of the mucus by donating an electron. The mucus from both snail species also showed reducing power; the activity of *P. canaliculata* mucus was significantly higher than the activity of the *L.*

*fulica* mucus. The reducing power is associated with the presence of reductants in the sample, which causes the reduction of Fe (III) to Fe (II) by donating a hydrogen ion. This reducing capacity reduces metal ions, especially ferrous ion which is pro-oxidant. In addition, our findings presented a different consequence from reducing power, where *L. fulica* mucus exhibited better total antioxidant activity than *P. canaliculata* mucus. These results indicated that antioxidant activity of the snail mucus was attributed to reducing capacity. This finding agrees with previous research which indicated that the extract from *L. fulica* mucus was particularly effective in reducing capacity, but it showed no DPPH free radical scavenging activity (*Kao et al., 2019*). This study is the first report for the antioxidant activity of mucus from *P. canaliculata*. The antioxidant structure and series of redox reactions in the activity influence the hydrogen or electron donation pattern of antioxidants (*Gupta et al., 2016*). Low molecular weight substances in snail mucus, such as vitamins C and E, phenolic compounds, uric acid, and uronic acid, generate antioxidant properties (*Wang et al., 2010*). The antioxidant ability of snail mucus is also related to the different peptides contained in each mucus, including amino acid composition, molecular mass, and hydrophobicity (*Wang et al., 2016*). Our findings suggest that mucus from *L. fulica* and *P. canaliculata* demonstrates the possibility of antioxidant activity as natural antioxidants.

In addition to having antioxidant ability, mucus from *L. fulica* and *P. canaliculata* also exhibited good anti-inflammatory activities with anti-NO production and anti-protein denaturation. NO is a proinflammatory mediator which has a role in several features of inflammatory diseases. A low level of NO production is necessary for maintaining normal body functions. In contrast, high concentrations of NO are due to the appearance of the inducible form of nitric oxide synthase (iNOS) in cells like macrophages and reacts with superoxide anions, generating peroxynitrite, which leads to the oxidation of low-density lipoprotein and causes cell apoptosis and subsequent inflammation (*Ohshima & Bartsch, 1994*; *Smith & Lassmann, 2002*). The results observed in this study showed that both mucus samples could decrease the production of NO in macrophage cells with concentration-dependent activity. The anti-inflammatory ability was correlated with the antioxidant activity because some antioxidants act as anti-inflammatory agents (*Traber, 2007*; *Gęgotek & Skrzydlewska, 2022*). Protein denaturation, when a protein loses its tertiary and secondary structures by external stress, heat, or strong acid or base (*Prasad, Yashwant & Aeri, 2013*), is related with the occurrence of the inflammation response. Major anti-inflammatory drugs, like NSAIDs, have been attributed to the capacity of anti-protein denaturation (*Alamgeer Uttra & Hasan, 2017*). In the present study, the *P. canaliculata* mucus exhibited a more potent anti-inflammatory effect than the *L. fulica* mucus.

Despite several applications of mucus from snail foot tissues, the histological and histochemical studies still need to be developed. Our previous study included ultrastructural studies on the snail foot tissue and the mucous cell localisation reported in *L. fulica* and *P. canaliculata*. The dorsal-ventral foot regions were observed, and it was found that their foot tissues were structurally similar. Still, the mucous cells of *L. fulica* and *P. canaliculata* differ in size and abundance (*Phrompanya et al., 2022*). Mucins, the primary component of snail mucus, are responsible for several biological activities. To evaluate the structure,

 

distribution, and function of mucus, detecting mucin types is necessary. In this study, the foot tissues of *L. fulica* and *P. canaliculata* were histochemically observed to classify mucin types in each foot region. Several studies utilize the standard histochemical classification of mucus secretions, especially the PAS and AB methods, since the different types of mucins contained in mucous cells can guide the functions of the mucous cells (*Di, 2012*) and the viscosity of mucus (*Grenon & Walker, 1980*).

The results of this study determined that the secretory product of the mucous cells in both species contained two types of mucins: one type produces a secretion containing neutral mucins, which stain positive with PAS, and acid mucins, which stain positive with AB. The ventral mucous cells of *L. fulica* presented many neutral and acid mucins which are related with their existence on land surfaces. The dorsal foot area of *P. canaliculata* secreted acid mucins in the same quantity as the dorsal foot area of *L. fulica*. Acid mucins create high viscosity of the mucus and play biological roles, such as reducing dehydration from their skin, or moisturising, and providing antibiotic and antipredator properties (*Jeong et al., 2001*). Neutral mucins play an essential role in the locomotion of snails by providing low-viscosity mucus (*Grenon & Walker, 1978*). According to this information, acid mucins are quite important in protective functions. It is highly viscous and a good lubricant because it is not easily hydrated and removed from the epithelium (*Faillard & Schauer, 1972*) and facilitates protection against bacteria (*Cao & Wang, 2009*). Therefore, mucus that contains a high proportion of acid mucins should provide better biological activity and cosmeceutical properties; however, the variation of mucin types could correlate with the age of the snails. The 3-month-old snail with a larger body size could produce more acid mucins than the 1- and 2-month-old snails (*Suwannapan et al., 2019*). The level of mucins is also affected by environmental factors, such as parasite infections (*Prociv, Spratt & Carlisle, 2000*) and metal exposure (*Londhe & Kamble, 2014*). Therefore, environmental disturbances may impact mucus production and its biological properties.

## CONCLUSIONS

Our research has shown that the mucus from *L. fulica* and *P. canaculata* possess antioxidant and anti-inflammatory properties. We have also studied the histochemical characteristics and distribution of mucins in the snail foot tissues. The mucus from *L. fulica* showed significant total antioxidant activity and was able to reduce NO production in LPS-induced macrophage cells. On the other hand, the mucus from *P. canaliculata* demonstrated notable reducing power and anti-protein denaturation properties. The dorsal foot region was the main area of acid mucin secretion; therefore, mucus from *L. fulica* and *P. canaliculata* should be collected from their dorsal foot region to obtain more effective mucus. The mucus found in freshwater snails (*P. canaliculata*) may have potential applications in medical and cosmetic products, akin to those of land snails (*L. fulica*).

### Abbreviations

| | |
|---|---|
| **AAE** | Ascorbic acid equivalent |
| **AB** | Alcian blue |
| **ABTS** | 2,2-azino-bis-(3-ethylbenzothiozoline-6-sulphonic acid) |

| | |
|---|---|
| **ANOVA** | One-way analysis of variance |
| **DMEM** | Dulbecco's Modified Eagle Medium |
| **DMSO** | Dimethylsulfoxide |
| **GAE** | Gallic acid equivalents |
| **H&E** | haematoxylin and eosin |
| **IC$_{50}$** | A half maximal inhibitory concentration |
| **LC$_{50}$** | the median lethal concentration |
| **LPS** | Lipopolysaccharide |
| **MTT** | 3-(4,5-Dimethylthiazol-2-yl)-2,5-diphenyltetrazolium bromide |
| **NO** | Nitric oxide |
| **PAS** | periodic acid-Schiff |
| **PBS** | phosphate-buffered saline |
| **QE** | Quercetin equivalent |
| **ROS** | Reactive oxygen species |
| **SD** | Standard deviation |

## ACKNOWLEDGEMENTS

We thank the Applied Microbiology laboratory unit, Department of Biology, Faculty of Science, Chiang Mai University for their help with the cell culture. The authors would like to thank Philip Martin Jones for the language review of this manuscript.

### Funding

This work was supported by the Human Resource Development in Science Project (Science Achievement Scholarship of Thailand, SAST), the Department of Biology, Faculty of Science, Chiang Mai University, the Ph.D.'s Degree Program in Biology (International Program), Faculty of Science, Chiang Mai University, and the Graduate School, Chiang Mai University. The funders had no role in study design, data collection and analysis, decision to publish, or preparation of the manuscript.

### Grant Disclosures

The following grant information was disclosed by the authors:
Human Resource Development in Science Project.
Department of Biology, Faculty of Science, Chiang Mai University.
Ph.D.'s Degree Program in Biology (International Program), Faculty of Science, Chiang Mai University.
Graduate School, Chiang Mai University.

### Competing Interests

The authors declare there are no competing interests.

## Author Contributions

- Phornphan Phrompanya performed the experiments, analyzed the data, prepared figures and/or tables, authored or reviewed drafts of the article, and approved the final draft.
- Narinnida Suriyaruean performed the experiments, analyzed the data, prepared figures and/or tables, and approved the final draft.
- Nattawadee Nantarat conceived and designed the experiments, authored or reviewed drafts of the article, and approved the final draft.
- Supap Saenphet analyzed the data, prepared figures and/or tables, and approved the final draft.
- Yingmanee Tragoolpua conceived and designed the experiments, authored or reviewed drafts of the article, and approved the final draft.
- Kanokporn Saenphet conceived and designed the experiments, authored or reviewed drafts of the article, and approved the final draft.

## Data Availability

The raw data are available in the Supplementary Files.

## Supplemental Information

Supplemental information for this article can be found online at http://dx.doi.org/10.7717/peerj.15827#supplemental-information.

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
