# Peer review of "Biological properties of mucus from land snails (Lissachatina fulica) and freshwater snails (Pomacea canaliculata) and histochemical study of mucous cells in their foot"

_PeerJ, doi:10.7717/peerj.15827_

## Round 0.1 · original submission · Minor Revisions

Kindly submit a revised version taking into consideration the feedback provided by the reviewers.

·

Basic reporting

Literature references, sufficient field background/context provided.

Comment:
- The introduction should include a section on free radicals and their negative effects on the body and the diseases resulting from their accumulation inside the body, as well as the role of natural products in mitigating them.

Experimental design

Methods described with sufficient detail & information to replicate.
Comment:
Animals and sample collection
- The authority granting a license or permission to collect samples should be mentioned.
- The authentication and identification of the collected samples should be mentioned.
- It is necessary to add a new section containing all used chemicals and reagents as well as their sources (Company name, city and country).
- A chemical characterization of the mucus should be carried out using HPLC-PDA-ESI-MS/MS, accompanied by molecular modeling, in order to link the chemical structure to the biological activities and to identify the chemical components responsible for these activities.

Validity of the findings

Conclusions are well stated, linked to original research question & limited to supporting results
Comment:
- The conclusion part should be supported by the results.

Additional comments

Abstract:
- The abstract is completely devoid of results. Some promising results should be mentioned (numbers or %).
Abbreviations:
- A list of abbreviations should be inserted by the end of the manuscript before references.
References:
- All scientific names of species should be written in italic fonts (e.g., Berberis orthobotrys; Pelteobagrus fulvidraco; Haliotis diversicolor; Seriola dumerili; Helix pomatia; Patella vulgata; Acmaea tessulata; Achatina fulica; Rhizopora mucronata; Bellamya bengalensis; Lissachatina fulica; Pomacea canaliculata; Coccinia grandis and Lissachatina fulica).
- The species name ''patella vulgata'' should be written as ''Patella vulgata''.
- The first letter in each word of journal names should be written in uppercase letters e.g.:
Journal of cosmetic dermatology
International journal for parasitology
Free radical biology and medicine
Analytical and quantitative cytology and histology
African journal of traditional, complementary, and alternative medicines
- References should be updated.

·

Basic reporting

In general, it is written in a clear language, it is only recommended to extend the summary of the methods part so that all the methods used are incorporated, as it is currently it only describes the objective but not the methodologies used.
The subject of study is relevant and of great interest on the evaluation of the properties of a widely used substance that is also associated with organisms that are of great interest due to their ecological impacts and commercial use.

Experimental design

The methodology used is adequate to achieve the proposed objectives.

Validity of the findings

The publication of the article with the suggested minor modifications is suggested, since it deals with a relevant topic in the description of the antioxidant and anti-inflammatory properties of snail mucus, which is of great interest in the cosmetics industry, and also provides information on a species that is shown as an alternative.

Additional comments

The literature was well referenced and relevant, although it is suggested to incorporate citations that validate some of the arguments proposed in the results section, which are detailed later.
It is necessary to restructure the text so that all methodological aspects are in the methods section. For example, lines 236-242, 246-249. It is suggested to eliminate isolated phrases that seem to re-incorporate methodological aspects and that do not contribute to the discussion of the topic: Line 315-306 “The phosphomolybdate method was used to determine the total in vitro antioxidant capacity of the mucus by transferring an electron”. In lines 324-326, 340- 342 it is necessary to include citation that validate the information presented (i.e Traber MG. Vitamin E regulatory mechanisms. Annu Rev Nutr 2007;27:347-62).
The figures were clear and informative enough. The title of figure 1 should be corrected, because the scientific name P. canaliculata is misspelled (P. canaliculate).
The tables are well structured, have relevant information and are presented in an orderly and clear manner. The title of table 1 and 2 should be corrected, because the scientific name P. canaliculata is misspelled (P. canaliculate).

---

## Round 0.2 · Minor Revisions

Thank you for taking the time to address all of the reviewers' comments. I have made some suggestions regarding grammatical and stylistic changes, which I have attached in a PDF file. Please take a moment to review them at your convenience and submit a final revised version.

---

## Round 0.3 · accepted · Accept

Thank you for taking care of the minor editorial concerns in the manuscript. It has been approved. The PeerJ editorial team will handle any additional details.